# An Improved Surgical Approach for Complete Interhemispheric Corpus Callosotomy Combined with Extended Frontoparietal Craniotomy in Mice

**DOI:** 10.3390/biomedicines11071782

**Published:** 2023-06-21

**Authors:** Ilja Jelisejevs, Jolanta Upite, Shivan Kalnins, Baiba Jansone

**Affiliations:** Department of Pharmacology, Faculty of Medicine, University of Latvia, LV-1586 Riga, Latvia; jelisejevs.ilja@gmail.com (I.J.); jolanta.upite@lu.lv (J.U.); shivan.kalnins@gmail.com (S.K.)

**Keywords:** corpus callosotomy, interhemispheric, craniotomy, mice, superior sinus retraction

## Abstract

Callosotomy is an invasive method that is used to study the role of interhemispheric functional connectivity in the brain. This surgical approach is technically demanding to perform in small laboratory animals, such as rodents, due to several methodological challenges. To date, there exist two main approaches for transecting the corpus callosum (CC) in rodents: trephine hole(s) or unilateral craniotomy, which cause damage to the cerebral cortex or the injury of large vessels, and may lead to intracranial hemorrhage and animal death. This study presents an improved surgical approach for complete corpus callosotomy in mice using an interhemispheric approach combined with bilateral and extended craniotomy across the midline. This study demonstrated that bilateral and extended craniotomy provided the visual space required for hemisphere and sinus retraction, thus keeping large blood vessels and surrounding brain structures intact under the surgical microscope using standardized surgical instruments. We also emphasized the importance of good post-operative care leading to an increase in overall animal survival following experimentation. This optimized surgical approach avoids extracallosal tissue and medium- to large-sized cerebral blood vessel damage in mice, which can provide higher study reproducibility/validity among animals when revealing the role of the CC in various neurological pathologies.

## 1. Introduction

The brain is an extremely fascinating and complex organ whose functionality is mediated by neuronal and structural connectivity patterns. The largest connective structure in the brain is the corpus callosum (CC) [1,2]. CC is a C-shaped white matter structure that connects the cerebral hemispheres and mediates information transfers between them [3]. Moreover, as a brain connector and coordinator, the CC has been strongly implicated in the pathogenesis of psychosis and in seizures of generalized, atonic, or unexplained natures (e.g., head trauma or alcohol withdrawal, instances with no obvious cause) [4,5]. It has been documented that, patients who experience multiple unprovoked focal and partial seizures, which could spread via the CC, may develop clinical epilepsy over time [5]. The transection of another brain structure, e.g., the fornix, is also described as being used to treat refractory temporal lobe epilepsy [6].

Corpus callosotomy, as a neurosurgical procedure, is a technique that involves partial or full separation of the cerebral hemispheres by visceral dissection of the CC. As a surgical method, it was first put into use to prevent the spread of contralateral and generalized seizures in the 1940s by the Chief of Neurosurgery at the University of Rochester, William Perrine Van Wagenen [7,8].

There are several clinically available callosotomy approaches, each offering advantages and disadvantages that depend on specific study goals, that are successful at breaking hemispheres and limiting the transmission of aberrant electrical activity. Such techniques include interhemispheric microsurgery, radiosurgery, endoscopic-assisted surgery, and laser surgery [9,10,11,12]. In human patients, corpus callosotomy is standardized; animal models present unique challenges and responses to extraneous damage.

Early reports on corpus callosotomy in animal studies, performed on dogs and cats, describe visual impairment if the medial cortex is damaged during surgery, and sensory or motor deficits if the rostral cortex is damaged [13,14]. Morphological, behavioral, and physiological functions of the CC have since been studied in different species of experimental animals [15]. Animal studies involving callosotomy in rodents, mainly using mice and rats, were originally designed to research the role of the CC in neurophysiological, behavioral, and motor functions [16,17,18]. Earlier described callosotomy studies conducted on small animals, like rodents, accessed the CC through either cranial window or trepanation hole surgery without interhemispheric access. These studies state that the CC can be either partially or completely dissected using these two approaches. To date, there are about 21 published papers that utilize corpus callosotomy in experimental studies in mice. The main weakness of these experimental studies is that techniques either sacrifice large parts of functional cerebral cortex (especially motor areas) to preserve the superior sinus or preserve the cerebral cortex while sacrificing the superior sinus [19,20,21,22,23]. Until now, advanced, detailed protocols or methodology descriptions for non-damaging CC surgery in rodents have not been available. Therefore, due to methodological limitations, corpus callosotomy has not frequently been used in rodent studies.

Currently, experimental studies in rodents have the chance to evolve due to advances in and the availability of technology. Examples include the development of fine surgical instruments, the widespread availability of high magnification/illumination surgical microscopes in animal facilities, and rodent-specific devices for neurosurgery such as stereotaxic frames. These developments provide a solid foundation to create a less invasive technique for CC dissection with very little impact on nearby blood vessels and cerebral structures. Therefore, this study aimed to provide a new approach for interhemispheric CC in mice by developing a detailed written and visual description of the inter-hemispheric surgical techniques employed.

The crucial difference between our approach and previously described methods is an inter-hemispheric callosotomy, combined with a retraction of the superior sinus and an extended craniotomy, all performed under surgical microscope in a stereotaxic frame. Accordingly, as opposed to other methods, the motor areas of one hemisphere are not sacrificed and congestive failure of blood circulation to the brain does not occur. We have provided detailed descriptions of a complete systematic protocol for corpus callosotomy surgery in mice that, ultimately, reduces extracallosal tissue damage.

## 2. Materials and Methods

### 2.1. Animals and Ethics Statement

For this study, six 10–12-week-old male C57Bl/6 mice (20–24 g) were used (Charles River Laboratories, Sulzfeld, Germany). Mice were grouped 5–6 per cage (polysulfone, W:395 × D:346 × H:213 mm) and housed within individually ventilated stainless-steel racks (Green Line IVC, GRM900 cages, Techniplast, Buguggiate, Italy).

Throughout the study, mice were maintained at the University of Latvia’s Faculty of Medicine Animal Care Facility, in accordance with standard experimental animal welfare regulations and under standard laboratory conditions (temperature, 23 ± 1 °C; humidity, 50–60%; 12 h day/night cycle, from 08:00 to 20:00; environmentally enriched cage). Bedding for each cage consisted of autoclaved aspen wood (Gold Aspen Classic B6, LBS-biotech, P.O. Box 431, London RH6 0UW, UK). For environmental enrichment, each cage contained: a polycarbonate tunnel (K3487, Techniplast, Buguggiate, Italy), crawl ball (K3329, Techniplast, Buguggiate, Italy), aspen blocks (1023005, LBS-biotech, London, UK), and aspen wood wool (1034005, LBS-biotech, London, UK). Animals received a standard rodent pellet diet (1321, Altromin, Lage, Germany) and had free access to filtered tap water, supplied ad libitum. Mice were habituated to the animal facility environment for 7 days before use in the experimental study.

The animal protocol for this study was approved by the Local Animal Welfare Committee of the University of Latvia and the Animal Ethics Committee of the Food and Veterinary Service, Riga, Latvia. All experimental procedures (i.e., study design, surgical manipulations, postoperative care, and humane endpoints) were performed in accordance with EU Directive 2010/63/EU and local laws and policies on the protection of animals used for scientific purposes. All efforts were made to minimize animal suffering and reduce the number of animals used.

### 2.2. Anesthesia Induction, Animal Positioning, and Surgery

For pain relief, thirty minutes before surgery, mice received a subcutaneous (s.c.) injection of 5 mg/kg carprofen (Carprofelican^®^ 50 mg/mL, Le Vet. B.V., Oudewater, The Netherlands) and an intraperitoneal (i.p.) injection of 0.1 mg/kg buprenorphine (Bupridex^®^ 0.1 mg/mL, PharmaSwiss, Zug, Switzerland).

To start anaesthesia, we used an anaesthesia induction chamber with console (World Precision Instruments, cat. no.: 502300, Sarasota, FL 34240, USA) and induced anaesthesia by using 0.3 L/min O_2_ and 0.7 L/min N_2_O in the presence of 4% isoflurane, maintaining the induced state with 2% isoflurane (Isoflurane, Veteasy, RWD, San Diego, CA 92121, USA). For the duration of surgery, we placed the mouse on a heating pad (RT-0501, Kent Scientific Corp, Torrington, CT 06790, USA) set to maintain an appropriate body temperature of 37.0 ± 0.1 °C. We placed the mouse in the prone position in a stereotaxic frame (World Precision Instruments, cat. no. EZ-B800, Sarasota, FL 34240, USA) and secured the head using ear and bite bar holders. We shaved the frontal and parietal parts of the mouse head fur with clippers (Med-Vet International, cat. no. 9757-300, Mettawa, IL 60045, USA) and disinfected the shaved skull skin and surrounding head fur with 70% alcohol, or equivalent skin disinfectant. We used a few drops of eye ointment (Corneregel^®^ 5%, PharmaSwiss, Zug, Switzerland) to prevent eye dryness during surgery. To assess the depth of anaesthesia, once the mouse was visibly asleep, awareness was checked using the toe pinch method. To initiate surgery, in brief, a 1 cm incision was made using fine-tipped scissors (ToughCut Fine Scissors, Code: 14058-09, Fine Science Tools GmbH, Heidelberg, Germany) in the rostra-caudal direction. To expose the skull, tissues were retracted using ophthalmic eyelid retracting hooks (World Precision Instruments, cat. no. 500369-G, Sarasota, FL 34240, USA) (see Figure 1a,b) and a dry surgical surface was created on the skull periosteum by careful scraping with a scalpel (World Precision Instruments, cat. no. 500240, Sarasota, FL 34240, USA). The whole procedure was performed under surgical microscope with 100× magnification (PZ-IV, World Precision Instruments, Sarasota, FL 34240, USA). Pulse oximetry, heartbeat, and respiratory rate were monitored using a physiological monitor (Physiosuite, Kent Scientific Corp, Torrington, CT 06790, USA).

### 2.3. Post-Operative Care

After surgery, all animals were housed individually under standardized conditions. The post-operative care period lasted for 1 week. The ambient temperature in the mice housing room was kept constant at 25 ± 0.5 °C. To increase survival rate after surgery, adequate food and water consumption was secured by providing mice with jelly-based food (D59494, Sniff Spezialdiäten GmbH, D-59494 Soest, Germany) and hydrogel (Clear H_2_O INC., Westbrook, ME 04092, USA). Jelly-based food was made fresh daily using 1 part food powder mixed with 3 parts water (ratio of 1:3). Jelly-based food, prepared fresh each day, and hydrogel were placed in Petri dishes and refilled daily. This was carried out in order to provide easy access to ad libitum water and food, particularly important for the first five days after surgery.

Each animal was carefully observed every day for any signs of pain. An analgesic effect was maintained by injecting buprenorphine (0.1 mg/kg, i.p.), every 8–12 h for the first two days after surgery. For four consecutive days after surgery, mice received injected carprofen (5 mg/kg, s.c.) at 24 h intervals. Post-surgery mice also received an injection of 20% glucose solution (0.5 mL, s.c., B. Braun, Munich, Germany) and Ringer’s Lactate solution (0.5 mL, s.c., B. Braun, Munich, Germany) once daily for 7 days. Animal weights were recorded before surgery and during the post-operative care period to assess surgery recovery. The overall health status (e.g., weight, the mouse grimace scale, coat condition, fast or difficulty breathing, tense and nervous upon handling, body function, environment, and locomotion) of all mice were monitored closely before surgery and during the post-operative care period.

### 2.4. Brain Sampling and Nissl Staining

To validate the developed surgical approach for complete interhemispheric corpus callosotomy, combined with extended frontoparietal craniotomy, two weeks after surgery, mice brains were isolated and freshly frozen with liquid nitrogen and then stored at −80 °C. When ready to stain, 20 nm thick slices were sectioned using a cryostat microtome (CM1950, Leica Biosystems, Munich, Germany) and mounted on gelatin-coated slides then left to dry completely. After sections were sufficiently dry, they were subjected to a staining sequence, as follows. Slides were placed in distilled water for 1 min, followed by 0.1% cresyl violet solution for 10 min (Sigma–Aldrich, St. Louis, MI, USA). Once removed from cresyl violet solution, slides were placed in distilled water for 3 min to rinse. Slides were then washed with 70% alcohol for 3 min followed by 95% alcohol for 2 min. After alcohol wash, slides were placed in xylene for 5 min and finished by mounting with DPX (100579, Sigma–Aldrich, St. Louis, MI, USA). After generation, slides were scanned for digital formatting (SCN400, Leica Systems, Munich, Germany).

## 3. Results

### 3.1. Craniotomy

The entrance for interhemispheric corpus callosotomy surgery was through an extended craniotomy. Using a fine diamond-tipped engraver (drill heads 0.6 mm/0.024 inch and 0.81 mm/0.032 inch, Cadillac, MI, USA), an elliptical cut, about 1.1 to 1.3 mm in diameter, was made from the bregma point to the lambda, the fusion point of the occipital bone with the parietal bones (Figure 1c). Care was taken if applying any drill pressure, as drilling through the bone is likely to cause damage to brain parenchyma; in general, the drill should require very little downward pressure for material removal. During this approach, damage-free results were achieved through very slow, measured drilling with frequent monitoring of depth progression and visual assessment of bone thickness—a skill that may require practice and consultation with other surgeons prior to competent and replicable surgery outcomes. When a remaining bone thickness of about 0.5 mm was reached, drilling was stopped.

To separate the superior sinus and dura mater from the created bone flap, a small amount of saline (NaCl 0.9%, B. Braun, Munich, Germany) was applied to the bone groove. The generated bone flap was gently pushed and pulled away from the skull (Figure 1d,e) and, for better regeneration after surgery, was placed in saline solution (NaCl 0.9%, B. Braun, Munich, Germany) during the remaining surgical procedure.

Bleeding, which may occasionally occur, was stopped by tamponade with Gelfoam^®^ (Pfizer, New York, NY, USA). Blood loss volume was monitored during surgery to determine the likely fate of the animal following surgery.

### 3.2. Callosotomy

Once visible, the dura mater surface was rinsed with saline and cleaned with cotton swabs (Figure 2a,b), then carefully pierced with micro tweezers (Code: 11231-30, Dumont, Fine Science Tools GmbH, D-69115, Heidelberg, Germany) to create flaps. The dura mater flaps were then placed aside (Figure 2c,d).

To begin the callosotomy procedure, the superior sinus was first retracted and repositioned with cotton swabs (Figure 3a–d). For CC dissection, a blunt-ended, short round micro hook (Code: 10064-14, Dumont, Fine Science Tools GmbH, D-69115, Heidelberg, Germany) was placed in the dorsoventral direction, 2.8 mm deep, inside the longitudinal fissure at the relative point nearest to the dura under the bregma (Figure 4a–d). All measurements were made according to metric gradation on the stereotaxic frame micromanipulator hand, to which the short round micro hook was attached, for high accuracy dissection.

The distance from the micro hook rod to the bregma was measured and, accordingly, the micro hook was moved 1.5 mm in the rostral-caudal direction from the dura under the bregma, targeting the frontal part of the CC (Figure 5a–d). The micro hook was then placed on the bregma and moved 3.5 mm in the caudal-rostral direction from the dura under the bregma, targeting the middle and rear parts of the CC (Figure 6a–d); this manipulation was controlled by the stereotaxic frame micromanipulator hand. For a full schematic overview of the corpus callosotomy technique instructions and micro hook placement coordinates, see Figure 7.

Surgery was concluded by anatomically repositioning the dura mater flap and then closing the cranial window by securing the bone flap to the skull using a mix of dental cement (DentAcryl^®^, 3M™ Scotchbond™ Adhesive Unit Dose 100 L-Pop, 41298, Riga, Latvia) and glue (Cyanoacrylate instant adhesive, 3M™, World Precision Instruments, Sarasota, FL 34240, USA) (Figure 8a,b). Dental cement was applied with a dental micro curette (WPS-725-34, World Precision Instruments, Sarasota, FL 34240, USA) around the bone flap and skull groove. The dental cement was kept in a semi-liquid state, by adding solvent when necessary, while attaching the bone flap to the skull and was dried completely before wound suturing.

Finally, wound edges were sutured with absorbable sutures (Prolene suture 6-0, Ethicon, B. Braun, Munich, Germany) and treated with iodophor solution (Betadine, EGIS, Riga, Latvia). Post-surgical mice received 0.5 mL of s.c. Ringer’s Lactate solution (B. Braun, Munich, Germany) and were placed in a recovery box (Recovery Chamber-V1200, Peco Services Ltd., Cumbria, UK) for 4 h with the temperature set to 28.5 ± 0.5 °C.

### 3.3. Observations during Post-Surgical Recovery and Nissl Staining Results

In this study, the general health and welfare of mice were monitored after a complete interhemispheric corpus callosotomy combined with an extended frontoparietal craniotomy. All of the operated animals survived the whole observation period of two weeks. Furthermore, a steady animal recovery, including food consumption and grooming behavior, was noticed within the first 24 h. No signs of complications in wound healing, including infection of the surgical site, were noted. A gradual weight gain was observed, and with good coat condition, and environment and locomotion activity were noted. The histological data of the mice’s brains—after complete interhemispheric corpus callosotomy combined with extended frontoparietal craniotomy—showed no brain damage and homogeneous outcomes in Nissl staining. The representative mouse brain coronal sections (see Figure 9b) indicated an absence of CC integrity along the midline, for the entire length, without any additional damage to the cortex.

## 4. Discussion

The goal of this study was to develop an improved approach for corpus callosotomy in adult mice combining an extended craniotomy with interhemispheric dissection access that is easily reproducible by other researchers and provides benefits over existing procedures. The techniques applied in our novel approach were in a direct effort to reduce blood vessel and/or extracallosal tissue damage following complete or partial dissection surgery. This novel approach to callosotomy in mice is made possible by advances in surgical and laboratory technology, as well as in materials, during the last decades.

In the published studies to date, there have been several techniques described for performing rodent callosotomy. These methods differ in how the CC is accessed and in the tools/methods used for transection. With respect to CC access, the two main methods include craniotomy and trephine hole. With unilateral or midline trephine hole access, even when using modern mitigating methods, drilling through soft tissues and skull material is not possible without introducing damage to the underlying structures [16,19,20,23,24]. Some of these structures could include portions of brain tissue and/or large blood vessels. The use of craniotomy is also common, with differences arising in craniectomy location. For example, one study used a unilateral craniotomy approach that separated the CC through intentional damage to the cerebral cortex [22]. In studies using intentional damage for separation, unilateral or bilateral cerebral cortex lesions were present in addition to damage of the following structures: fornix, hippocampus, cerebellum, colliculi, septal nucleus, and mesencephalon [16,19,20,21,22,23,24,25]. The impacts of damage to these structures are unpredictable and varied. All play important roles in brain function and process integration. For example, the fornix and hippocampus are key in cognition and memory [26,27]. The cerebellum has a well-known role in locomotor control and learning [28]. The colliculi are a vital integrator of visual, somatosensory, and auditory information when initiating motor commands [29]. In close spatial and functional proximity to the hippocampus and hypothalamus, the septal nucleus has a major role in emotional, behavioural, and feeding processes [30]. Other researchers made use of a bilateral craniotomy without an extension through the brain midline, which allowed moderately constricted interhemispheric access for corpus callosotomy that reduced, but did not eliminate, damage to other proximate tissues and large blood vessels [25,31,32]. While also using bilateral craniotomy for CC access, the main difference between previous studies and the approach in our study is the extension across the brain midline. Importantly, this extension increased surgical manipulation space, resulting in better visual control under surgical microscope and further allowed for hemisphere and sinus retraction.

A review of previous studies involving partial or complete transection of the CC indicates notable damage to important brain structures. These damages were partly caused by unoptimized surgical equipment. The materials used to conduct the corpus callosotomy procedure and the transection path taken can also have direct impacts on surgery quality. Self-made tools have historically been used in past studies, with only brief mentions of construction methods, or tools from other professions have been adapted for use. For example, some studies described using manufactured dental tools during transection [20,21]. Another study introduced the novel application of a cutting electrode that necessitated the sacrifice of tissues to operate [25]. Self-made, flexible transection instruments, made from metal gauge wire or silk suture, have also been described in early studies [16,23]. Additionally, self-made sharpened or blunted knives have seen use in more recent transection procedures [19,20,22,24]. Based on our current experience with this procedure, we believe that excessive brain damage and surgical imprecision could result from the use of sharpened instruments, electrocoagulation to stop haemorrhage, and/or the use of metal gauge wires or needles. For the dissection of the CC with minimal damage to the extracallosal tissues, blunt instruments used on soft tissues are best.

For optimal healing and post-operative function in mice undergoing full or partial corpus callosotomy, the preservation of brain structures is of the highest importance; the degree and combination of injuries from improper surgical instrument choice could lead to unpredictable and/or nonreproducible results after surgery.

During the literature review, it was observed that corpus callosotomy studies in rodents from the 2010s to now mostly refer to methodology from the 1980s or 1990s, wherein lab-constructed wire gauge knives were used during dissection [31,32]. The availability of innovative biomedical laboratory instruments, surgical tools, and maintenance devices has dramatically and globally expanded in recent decades. To ensure reproducibility between research labs and from study to study, standard tools that are easily accessible to any researcher should be used whenever possible. In situations where standard tools cannot be used, the exact specifications of lab-made tools should be provided and, if possible, alternate solutions suggested. To adhere to this standard, the current study precisely states, within the Materials and Methods and Results sections, all identifying and commercial information required for independent researcher acquisition, including tool names and manufacturer codes. In the context of damage-mitigating CC dissection in mice, commercially available fine surgical instruments, equipment, and micromanipulator hands are vital in damage-free and precise transection surgery. Furthermore, maintenance equipment such as heating pads and recovery boxes are significant contributors to the long-term healing success of mice receiving CC surgery.

In addition to surgical tool considerations, to increase CC access and prevent vessel damage during dissection, superior sinus repositioning is a crucial stage for damage-free success. Of the studies published that performed callosotomy transection under the superior sinus in the longitudinal fissure in mice, none provide information on superior sinus repositioning, retraction materials, or techniques [16,21,23]. One study using rats makes mention of superior sinus repositioning; however, how it is performed is not stated in the method descriptions, and structural impairments to the brain were evident when observing the study images [25]. Iatrogenic damage to the superior sinus, a result of its fragile structure, can be avoided using high-magnification surgical microscopes, gentle manipulation during repositioning, and advanced haemostasis solutions, for example, Gelfoam^®^. Iatrogenic damage could cause a little bleeding, that can be treated by tamponade or, the damage can cause massive, unstoppable haemorrhages requiring exclusion and euthanasia of the animal. The incorrect assessment of the superior sinus condition may subsequently lead to intracranial haemorrhage or insufficiency of the superior sinus with associated cerebral edema [33]. Accordingly, utmost care in the manipulation and the ultimate avoidance of superior sinus damage should be taken. Proper post-operative care following manipulation is also highly important in surgery success.

There are several practical refinement approaches that can be easily applied to reduce animal suffering after surgery, increase the rate and time of recovery, and increase overall survival rates thereby improving overall study quality. In addition, optimization of post-operative care will directly contribute to a reduction in animal use and mortality per study. Refinements include hypothermia mitigation, securing appropriate food and water sources, and provision of analgesics for pain. Of note, hypothermia caused by surgery and anaesthesia slows recovery and increases the risk of post-operative mortality [34]. In our study, hypothermia risk was reduced by maintaining optimal body temperature not only with a heating pad during the surgery but, importantly, following surgery by placing the mouse for 4 h in an automated recovery box set to 28.5 °C. Following this initial recovery period, when kept in the animal housing room, temperatures were always no lower than 25 °C. It has additionally been observed that after brain manipulations, such as filamentous middle cerebral artery occlusion (fMCAo), there is secondary weight loss attributable to the disruption of systemic processes combined with low food and water consumption during post-operative recovery. In the referred study, proper post-operative consumption care was associated with a reduction in 14-day mortality by up to 75%. Furthermore, given post-operative weakness, there is a significant chance that hypoglycaemia and/or dehydration may develop. To avoid this consequence in our study, during the crucial 24 h period directly after, and one week following, surgery, s.c. injections of 20% glucose solution and Ringer’s Lactate solution were provided. Sufficient fluid and nutritional intake were secured using easily consumable products placed in easily accessed locations; for instance, a jelly-based food made of standard rodent food powder mixed with water, placed in a Petri dish on the cage floor, was used until mice regained at least 85% of their initial body weight. After weight gain, jelly-based food in the Petri dish was gradually replaced with water-soaked rodent pellets and later by dry rodent pellets. Hydrogel, to support fluid intake, was also provided in a separate Petri dish on the cage floor during post-operative recovery. Be aware that a week after surgery, mice may consume more food and water overall, compensating for their reduced intake in the previous period [35]. Altogether, applying the abovementioned steps will considerably increase animal survival both immediately following surgery and throughout the course of extended studies.

There are limitations to this method. While methods to avoid damage to the superior sinus have been addressed, unfortunately, damaging the small pericallosal blood vessels, due to their anatomy and the process of separating the CC [36], is unavoidable. This results in minor haemorrhage and requires a rapid haemostasis response. The importance of the pericallosal blood vessels during CC dissection is unknown and requires further investigation.

Another limitation, of any surgery requiring the manipulation of dura mater in rodents, is the complete technical inability to ensure the closure of the dura mater such that there is no leakage of cerebrospinal fluid into the cranial cavity. Massive leakage may cause physical and health impacts on the affected animal. It would be highly important to have an available technique to close the dura mater; however, even with modern surgical options, it is dramatically challenging to perform on a small scale, such as in mice. It is significant to note that there is a link between the severity of traumatic brain damage and gradually increasing reactive astrogliosis, which may lead to the induction of epileptogenic foci [37].

Quality control after callosotomy surgery is also important. It would be ideal to have live imaging, such as MRI (Magnetic resonance imaging), available to determine adequate brain function following CC dissection. However, the cost of, space for, and availability of MRI machines do not make MRI a universally feasible means of surgery validation in experimental studies. In the interim, a post-mortem method like Nissl staining, while far from ideal, is a simple, cost-effective, and easily reproducible histological technique to visually demonstrate the dissection of the CC while simultaneously highlighting instances of rupture or damage. As such, we suggest including Nissl staining to confirm the absence of CC integrity and the absence of damage to unintended tissues.

## 5. Conclusions

Callosotomy is a complex and challenging surgery in small rodents. The existing methodologies do not avoid extracallosal tissue damage and/or medium-to-large blood vessel damage following the complete or partial dissection of the CC in mice. This paper describes an improved approach to corpus callosotomy in mice that combines an extended craniotomy with the interhemispheric dissection of the CC, thus playing a vital role in the extracallosal tissue’s damage-free dissection. The repositioning of the superior sinus during surgery is another vital care taken to minimize possible trauma to blood vessels. This small-scale surgery is advised to be performed under surgical microscope using commercially available materials and tools. We would like to highlight that for successful studies, a well-considered post-operative maintenance care system is crucial in order to increase overall survival rates. This improved surgical approach to corpus callosotomy can be easily adapted for use in both complete or partial dissection, and in younger or older mice (following the appropriate bregma coordinates for subject mouse age), to study the role of the CC in brain diseases.

## Figures and Tables

**Figure 1 biomedicines-11-01782-f001:**
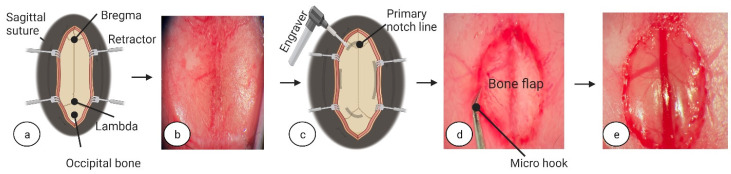
**Bone flap creation.** (**a**) Visual explanation for (**b**) mice skull through the microscope. (**c**) Using a diamond-tipped engraver, we drilled an ellipsoid notch from the bregma point to the lambda (about 8 mm). Being careful to avoid pressure, we drilled the groove to a bone thickness of ~0.5 mm. (**d**) With a micro hook tip, we picked up the bone flap and carefully lifted it. For better results when separating dura mater from the bone flap, we applied a few drops of saline inside the skull groove. (**e**) Visualization of superior sinus and blood vessels after bone flap removal. Image (**a**,**c**) was created at BioRender.com, accessed on 5 June 2022.

**Figure 2 biomedicines-11-01782-f002:**
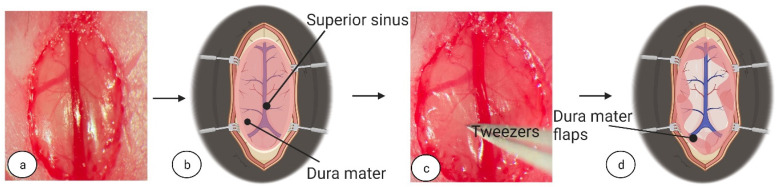
**Dura mater flap creation.** (**a**) After removing the bone flap, there was visible dura mater and superior sinus. (**b**) Visual explanation for (**a**). (**c**) We carefully pinched dura mater with micro tweezers, creating flaps, and placed aside as in (**d**). Image (**b**,**d**) created at BioRender.com, accessed on 5 June 2022.

**Figure 3 biomedicines-11-01782-f003:**
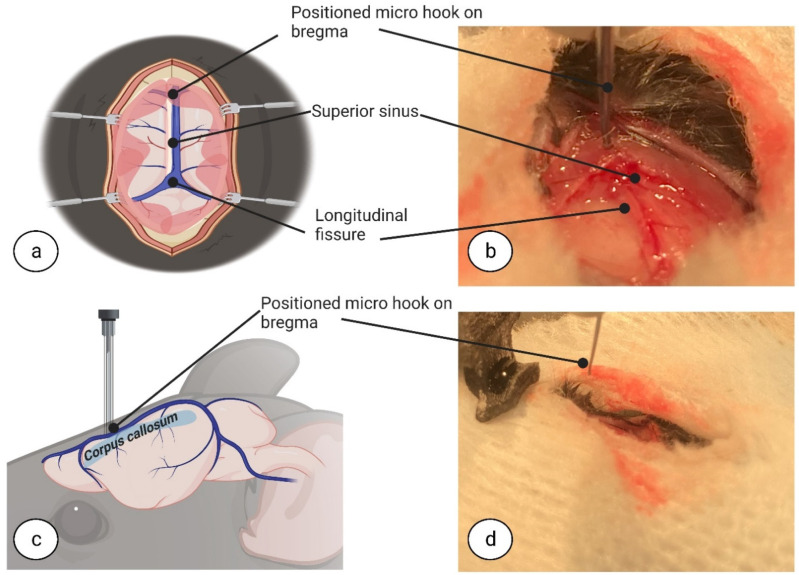
**Micro hook positioning on bregma.** The superior sinus was gently retracted and repositioned using a cotton swab followed by positioning of the micro hook, which had been placed, in a micro-manipulator, on the bregma between the two hemispheres within the longitudinal fissure. Top view (**a**,**b**). Side view (**c**,**d**). Image (**a**,**c**) created at BioRender.com, accessed on 5 June 2022.

**Figure 4 biomedicines-11-01782-f004:**
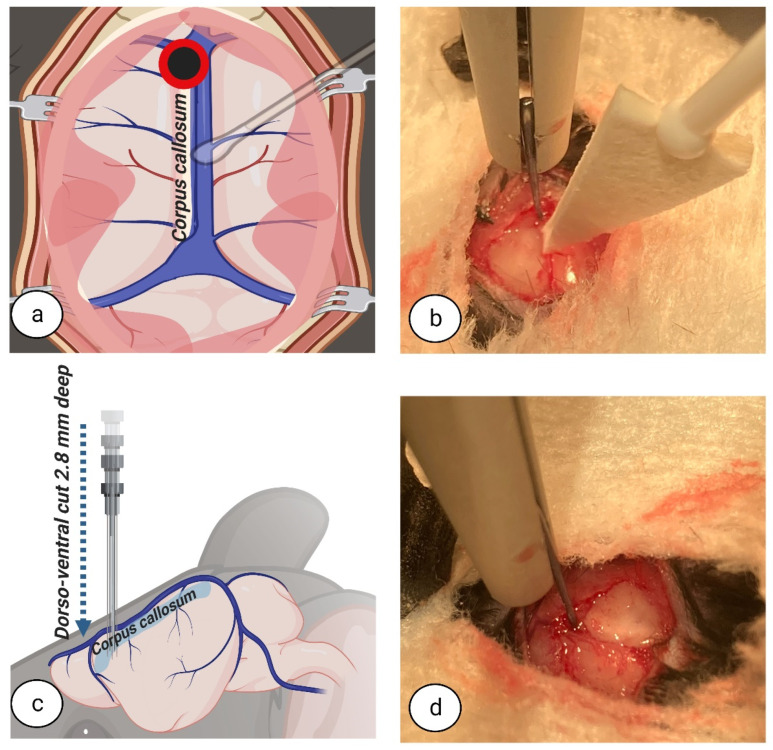
**Dorsoventral dissection.** The blunt end of the micro hook was placed in the dorsoventral direction, 2.8 mm deep, inside the longitudinal fissure at the relative point nearest to the dura under the bregma. Top view (**a**,**b**). Side view (**c**,**d**). Image (**a**,**c**) created at BioRender.com, accessed on 5 June 2022.

**Figure 5 biomedicines-11-01782-f005:**
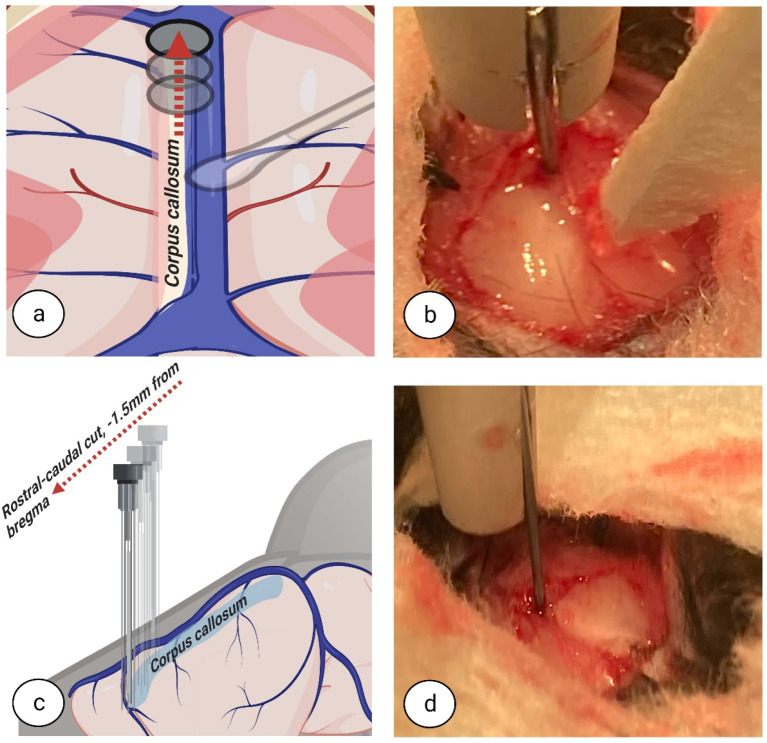
**Rostral-caudal dissection.** The distance from the micro hook rod and bregma was measured (1.5 mm) and then moved in the rostral-caudal direction, targeting the frontal part of the corpus callosum. Top view (**a**,**b**). Side view (**c**,**d**). Image (**a**,**c**) created at BioRender.com, accessed on 5 June 2022.

**Figure 6 biomedicines-11-01782-f006:**
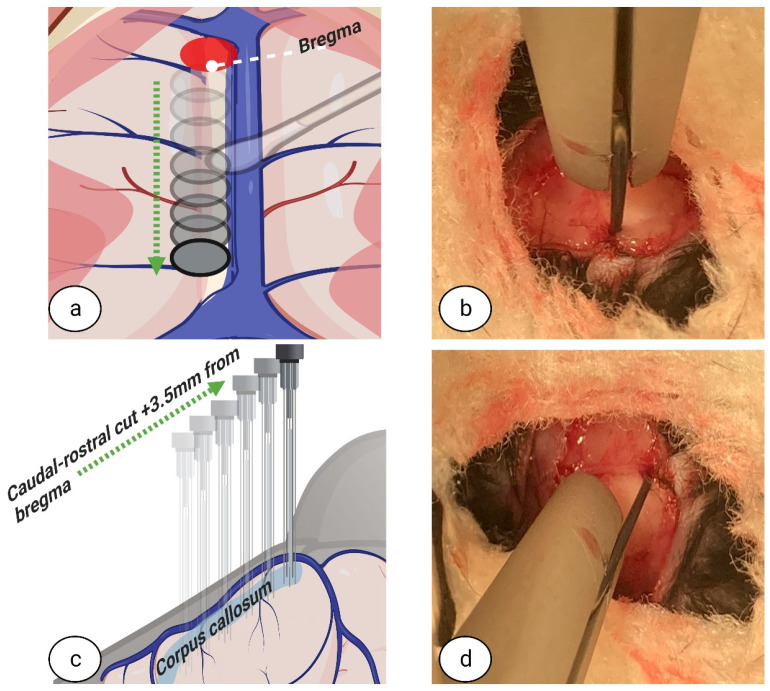
**Caudal-rostral dissection.** The micro hook was placed on the bregma and moved in the caudal-rostral direction 3.5 mm from the bregma, to target the middle and rear parts of the corpus callosum. Top view (**a**,**b**). Side view (**c**,**d**). Image (**b**,**d**) created at BioRender.com, accessed on 5 June 2022.

**Figure 7 biomedicines-11-01782-f007:**
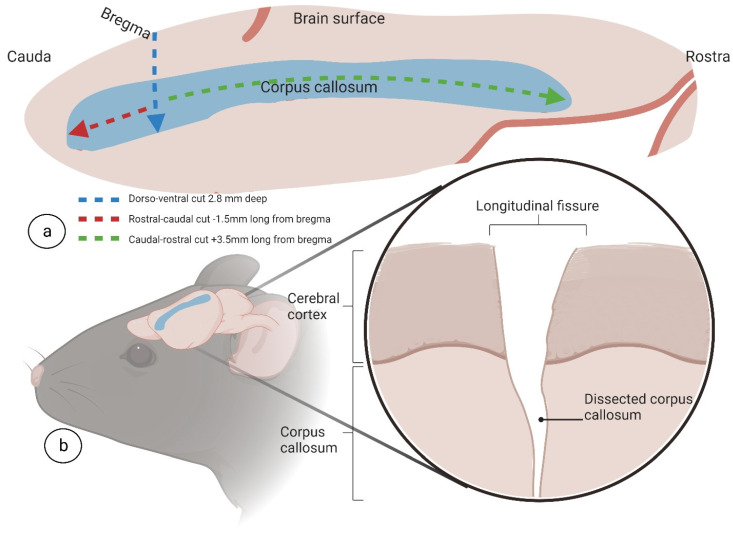
**Dissection schematic.** A schematic overview of the cuts required for complete corpus callosotomy, including coordinates relative to bregma. A view from the top (**a**) and inset, magnified coronal view (**b**). Image created at BioRender.com, accessed on 5 June 2022.

**Figure 8 biomedicines-11-01782-f008:**
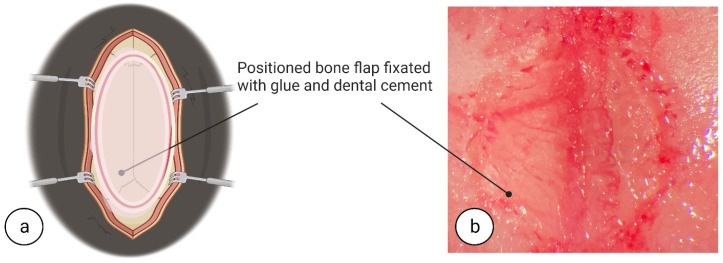
**Post-surgical fixation of the bone flap.** Surgery was concluded by anatomically positioning the bone flap and securing it with dental cement that had been kept in a semi-liquid state via addition of solvent when required (**a**,**b**). Dental cement was applied to the bone flap/skull groove using a dental micro-curette. Dental cement should dry completely prior to wound suturing. Image (**a**) created at BioRender.com, accessed on 5 June 2022.

**Figure 9 biomedicines-11-01782-f009:**
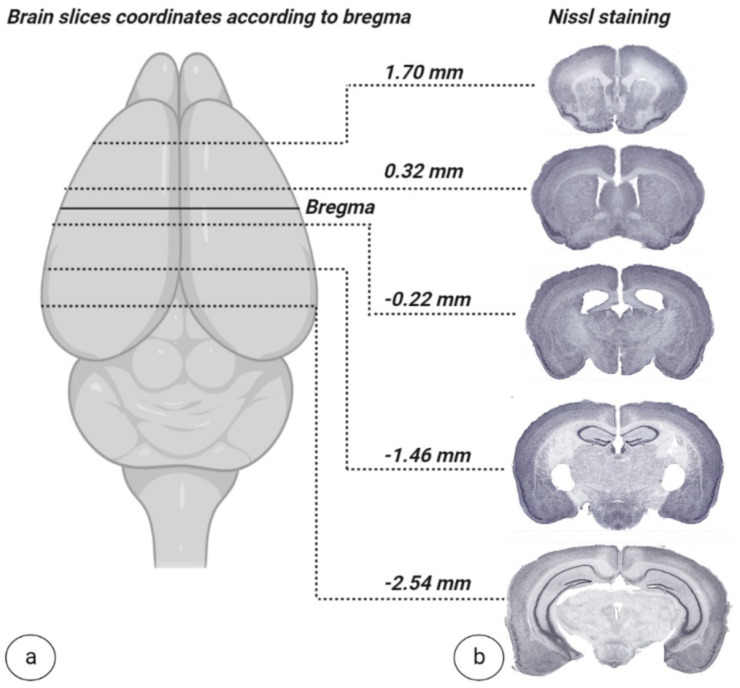
**Co-ordinate specific Nissl-stained sections.** (**a**) Visual representation of coordinates relative to bregma. (**b**) Nissl-stained sections of callosotomized mouse brain. Image (**a**) created at BioRender.com, accessed on 5 June 2022.

## Data Availability

The authors declare that all data supporting the methodology and findings of this study are available within the article.

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
