# Peer review of "An Improved Surgical Approach for Complete Interhemispheric Corpus Callosotomy Combined with Extended Frontoparietal Craniotomy in Mice"

_biomedicines, 2023, doi:10.3390/biomedicines11071782_

Round 1

Reviewer 1 Report

The authors present a detailed surgical technique and methodology about the procedure of callosotomy in mice. They kept there approach minimally invasive in order to minimize morbidity and mortality. 

This paper is very instructive for the lab personell who needs to perform this procedure, the included  illustrations are very good.

However, the topic is very specific and the main focus is on the methodology of an improved but very special surgical procedure. This special topic may not be of interest for the majority of the readers.

Well written, only minor editing necessary

Author Response

Answer to Reviewer #1:

  • Reviewer #1 Point 1: Well written, only minor editing necessary.

Response: Thank you for your suggestion and recognition. The whole manuscript was once again carefully redacted (track-changes indicate the improvements) by Shivan Kalnins who is a Canadian-based, native English speaker and has both degree in Animal Biology, and a graduate certificate in Technical Writing from Algonquin College (Ottawa, ON). Particular focus on experimental design. The graduate certification further developed written communications, including principles of editing, such that we feel her skills are sufficient for final English language editing.

Reviewer 2 Report

The authors described an improved surgical approach for complete corpus callosotomy in mice using an interhemispheric combined with extended frontoparietal craniotomy across the midline. They found that bilateral and extended craniotomy provided the visual space required for hemisphere and sinus retraction and thus keeping large blood vessels and surrounding brain structures intact. So, they concluded that this model avoids extra-callosal tissue and cerebral blood vessel damage and provide higher reproducibility of study among animals.

Overall, the procedures are clearly described. This manuscript also contributes to the research community of corpus callosotomy and put forward several vital points for the surgical procedures.

In the meanwhile, I have the following comments could improve the manuscript.

1.      The authors mentioned that six male mice were used in this study. However, no details of these six mice condition (e.g. mortality, model successful or not, brain cortex damage, recovery condition, incision infection, etc.) regarding procedure were described in the manuscript.

2.      In the “3.1. Craniotomy” paragraph, “Care was taken when applying any pressure, as drilling through the bone would likely cause damage to brain parenchyma. When a remaining bone thickness of about 0.5 mm was reached, drilling was stopped”. How can the surgeon know the remaining bone thickness is around 0.5 mm during the drilling and stop it? Craniotomy is the step normally cause cortex damage when the drill broken the skull and the dura. So, this step has no difference compared with other craniotomy, and has not shown any advantages to other procedures regarding brain cortex damage during craniotomy.

3.      In the “3.2. Callosotomy” paragraph, “For CC dissection, a blunt-ended, short round micro hook was placed in the dorsoventral direction, 2.8 mm deep, inside the longitudinal fissure at the nearest point to the bregma”. So, the authors should clarify that 2.8 mm deep is relative to the bregma (skull) or the dura under the bregma (skull), since skull has certain thickness.

4.      The authors should describe how to control the 2.8 mm depth during the micro hook was moved 1.5 mm in the rostral-caudal direction and 3.5 mm in the caudal-rostral direction from the bregma, since the surface of the brain is a curve instead of a straight line as shown in the Figure 3C.

Author Response

Answers to Reviewer #2:

  • Reviewer #2 Point 1: The authors mentioned that six male mice were used in this study. However, no details of these six mice condition (e.g. mortality, model successful or not, brain cortex damage, recovery condition, incision infection, etc.) regarding procedure were described in the manuscript.

Response: We would like to thank the reviewer for highlighting this point. Based on the recommendation of reviewer #2, in the manuscript, ID: biomedicines-2384841, we updated and added text on how we monitored the general health status of the mice before and after the surgery.

Please see the new text added in the revised manuscript section – 2.3. Post-operative care (Lines 163 – 166):

“The overall health status (e.g., weight, the mouse grimace scale, coat condition, fast or difficulty breathing, tense and nervous on handling, body function, environment, and locomotion) of all mice were monitored closely before surgery and during the post-operative care period.”

We would like to point out that we have made an additional change in the section “Results” by renaming the chapter title for section 3.3. - "Observations during post-surgical recovery and Nissl staining results" to improve the overall quality of the manuscript (Lines 281-288).

“In this study, we monitored the impact of a complete interhemispheric corpus callosotomy combined with an extended frontoparietal craniotomy on the general health and welfare of mice. All the operated animals survived the whole observation period of two weeks. We observed a steady animal recovery as food consumption, and grooming behavior were noticed within the first 48 hours. No signs of complications in wound healing, including infection of the surgical site, were noted. A gradual weight gain was observed, well coat condition, environment and locomotion activity were noted.”

  • Reviewer #2 Point 2: In the “3.1. Craniotomy” paragraph, “Care was taken when applying any pressure, as drilling through the bone would likely cause damage to brain parenchyma. When a remaining bone thickness of about 0.5 mm was reached, drilling was stopped”. How can the surgeon know the remaining bone thickness is around 0.5 mm during the drilling and stop it? Craniotomy is the step normally cause cortex damage when the drill broken the skull and the dura. So, this step has no difference compared with other craniotomy, and has not shown any advantages to other procedures regarding brain cortex damage during craniotomy.

Response: Overall, we agree to the review comment. To emphasize the importance of making this procedure very carefully, we made an additions to the text, see also in the manuscript (Lines 188-193)

“in general, the drill should require very little downward pressure for material removal. During this approach, damage-free results were achieved through very slow, measured drilling with frequent monitoring of depth progression and visual assessment of bone thickness – a skill that may require practice and consultation with other surgeons prior to competent and replicable surgery outcomes”.

  • Reviewer #2 Point 3: In the “3.2. Callosotomy” paragraph, “For CC dissection, a blunt-ended, short round micro hook was placed in the dorsoventral direction, 2.8 mm deep, inside the longitudinal fissure at the nearest point to the bregma”. So, the authors should clarify that 2.8 mm deep is relative to the bregma (skull) or the dura under the bregma (skull), since skull has certain thickness.

Response: Thank you for pointing this out, we have clarified in the text that 2.8 mm deep is relative to the dura under the bregma (Lines 219-220 and 256-257).

  • Reviewer #2 Point 4: The authors should describe how to control the 2.8 mm depth during the micro hook was moved 1.5 mm in the rostral-caudal direction and 3.5 mm in the caudal-rostral direction from the bregma, since the surface of the brain is a curve instead of a straight line as shown in the Figure 3C.

Response: Thank you for the suggestion to add an additional clarification. We extended the sentence by inserting the text: this manipulation was controlled by the stereotaxic frame micromanipulator hand’ (Lines 227-228).
